# Personalized Early-Warning Signals during Progression of Human Coronary Atherosclerosis by Landscape Dynamic Network Biomarker

**DOI:** 10.3390/genes11060676

**Published:** 2020-06-20

**Authors:** Jing Ge, Chenxi Song, Chengming Zhang, Xiaoping Liu, Jingzhou Chen, Kefei Dou, Luonan Chen

**Affiliations:** 1Shanghai Institute of Biochemistry and Cell Biology, Center for Excellence in Molecular Cell Science, Chinese Academy of Sciences, Shanghai 200031, China; gejing@sibs.ac.cn (J.G.); zhangchengming2017@sibcb.ac.cn (C.Z.); xpliu@sdu.edu.cn (X.L.); 2State Key Laboratory of Cardiovascular Disease, Department of Cardiology, Fuwai Hospital, Chinese Academy of Medical Sciences, National Center for Cardiovascular Diseases & Peking Union Medical College, Beijing 100037, China; songcxfw@126.com (C.S.); chendragon1976@aliyun.com (J.C.); 3Key Laboratory of Systems Biology, Hangzhou Institute for Advanced Study, University of Chinese Academy of Sciences, Chinese Academy of Sciences, Hangzhou 310024, China; 4School of Mathematics and Statistics, Shandong University at Weihai, Weihai 264209, China; 5Center for Excellence in Animal Evolution and Genetics, Chinese Academy of Sciences, Kunming 650223, China; 6School of Life Science and Technology, ShanghaiTech University, 100 Haike Road, Shanghai 201210, China

**Keywords:** single-sample network, landscape dynamic network biomarkers (l-DNB), tipping point, coronary atherosclerosis, myocardial infarction

## Abstract

Coronary atherosclerosis is one of the major factors causing cardiovascular diseases. However, identifying the tipping point (predisease state of disease) and detecting early-warning signals of human coronary atherosclerosis for individual patients are still great challenges. The landscape dynamic network biomarkers (l-DNB) methodology is based on the theory of dynamic network biomarkers (DNBs), and can use only one-sample omics data to identify the tipping point of complex diseases, such as coronary atherosclerosis. Based on the l-DNB methodology, by using the metabolomics data of plasma of patients with coronary atherosclerosis at different stages, we accurately detected the early-warning signals of each patient. Moreover, we also discovered a group of dynamic network biomarkers (DNBs) which play key roles in driving the progression of the disease. Our study provides a new insight into the individualized early diagnosis of coronary atherosclerosis and may contribute to the development of personalized medicine.

## 1. Introduction

Cardiovascular diseases (CVDs) are the leading cause of mortality in the world, accounting for almost one-third of deaths worldwide [1]. Among the deaths caused by CVDs, ischemic heart disease accounted for 42.5%, and the primary cause is coronary atherosclerosis [1,2]. Coronary atherosclerosis begins with intimal hyperplasia near the bifurcation of the coronary artery, and then causes stenosis or obstruction of the vascular lumen, eventually leading to fatal cardiovascular events such as myocardial infarction [3,4]. Clinically, visual stenosis of 50–70% by coronary angiography was defined as the critical lesion of coronary atherosclerosis, which is a common pathological phenotype of the disease [5,6]. Although the degree of coronary stenosis in patients with critical lesions is similar, the prognosis varies greatly. Some patients remained in a stable state for a long time, while others deteriorated rapidly, leading to major adverse cardiovascular events. Therefore, it is extremely important to develop novel methods and biomarkers to make risk stratification more accurate and identify patients at high risk of adverse events as early as possible.

In recent years, systems biology approaches have been widely applied in the studies of cardiovascular-related diseases [7,8,9,10]. The progression of coronary atherosclerosis is a chronic and nonlinear process, which involves complicated dynamic regulations in biomolecular networks. For human coronary atherosclerosis, the progression process can be generally divided into three stages—normal state, predisease state (i.e., tipping point) and disease state (major adverse cardiovascular events, e.g., myocardial infarction). The system will rapidly deteriorate into an irreversible disease state just after the tipping point. Therefore, for early prediction and prevention of adverse cardiovascular events, it is critical to identify the predisease state of the coronary atherosclerosis, especially accurate to the individual level.

Based on the dynamic evolution characteristics of complex disease, we developed the theory of dynamic network biomarkers (DNBs) to identify the tipping point and leading molecular networks during the progression of a complex disease [11], which has been successfully applied in many diseases [12,13]. However, to detect the tipping point of each patient, the original DNB method requires multiple samples, which are not available generally for individual patients in clinical practices. In addition, it is not trivial to computationally determine DNB members and DNB module size. To solve those problems, the landscape DNB (l-DNB) method was developed to identify the tipping point of diseases from a single sample [14]. Based on the three criteria of traditional DNB theory and single-sample network, the l-DNB method can evaluate the local DNB score for molecule by molecule in a sample, and then compile all of the local DNB scores into a landscape of this sample. Therefore, the l-DNB method can be applied to identify the tipping point of human coronary atherosclerosis at the individual level.

In this study, we applied the local DNB score of molecules in each sample by using metabolomics data from different stages of coronary atherosclerosis patients. We not only identified the tipping point of the disease, but also predicted the criticality (i.e., early warning signals) of each patient and exactly identified the patients at high risk of adverse cardiovascular events verified by the follow-up information of these patients over the years. Furthermore, we also discovered a group of molecular network biomarkers. Evaluating them as a network by second-order statistics (e.g., correlation and deviation of these molecules) but not by the traditional first-order statistics (e.g., their average values) to assess the disease status, will provide new insights into the discovery of novel network biomarkers.

## 2. Methods

### 2.1. Patient Information and Study Design

Patients in this study received selective coronary angiography between June 2011 and March 2015 in Fuwai Hospital (Chinese Academy of Medical Sciences, Beijing, China). We excluded the patients with rheumatic heart disease, cardiomyopathy and other organic heart diseases, and also excluded the patients with severe liver and renal dysfunction, severe infectious diseases, malignant tumors, immune system diseases, connective tissue diseases, hyperthyroidism, Cushing syndrome and other metabolic diseases. Informed consent was obtained from all study participants. This study was performed complying with the Declaration of Helsinki and was approved by the Ethics Committee of Fuwai Hospital. 

Forty-eight patients were divided into three groups according to their coronary angiographic results. Nineteen patients with stenosis of coronary arteries < 20% were regarded as the control group (i.e., reference samples). Among them, about 90% of people had no coronary stenosis diagnosed by coronary angiography. Fifteen age- and sex-matched patients with stenosis of the coronary arteries between 50% and 70% were sorted into the group of Stage A. In group of Stage B, 14 patients were diagnosed with acute myocardial infarction (AMI). The follow-up information of the patients was collected in August 2019. One patient (ID: XJ608) in Stage A group received revascularization in September 2018 due to cardiovascular disease. The study design is shown in Figure 1a. 

### 2.2. Serum Collection and Preparation

Fasting plasma samples (4 mL) were collected before coronary angiography in heparinized tubes and centrifuged within one hour of collection (4 °C, 10 min at 2300 rpm). The plasma sample was then separated into aliquots and immediately frozen at −80 °C for metabolomics analysis. 

A total of 400 μL of extraction solvent (V methanol: V acetonitrile = 1:1) was added to 100 μL of plasma thawed at 4 °C, followed by incubation for 1 h at −20 °C to precipitate proteins. The mixture was then centrifuged at 12,000 rpm for 15 min at 4 °C, and the supernatant (425 μL) was transferred into a new eppendorf tube. After drying and re-dissolution in acetonitrile:water (1:1), 60 μL of supernatant was transferred into a 2 mL glass vial for liquid chromatograph-mass spectrometer (LC-MS). Quality control samples were prepared by pooling 10 μL supernatant from each sample. 

### 2.3. Metabolomics Study

The metabolomics of plasma was performed on a UHPLC system (1290, Agilent Technologies, Santa Clara, CA, USA) with a UPLC BEH Amide column (1.7 μm 2.1 ×* 100 mm, Waters) coupled to TripleTOF 6600 (Q-TOF, AB Sciex, Foster City, CA, USA). MS raw data files (wiff) were converted to the mzXML format by using ProteoWizard software (AB Sciex, Foster City, CA, USA). Retention time alignment, peak discrimination, data filtering, alignment and matching were carried out by using R package XCMS (version 3.2), which generated a data matrix that consisted of the retention time (RT), mass-to-charge ratio (*m*/*z*) values, and peak intensity. Then, CAMERA R package was used for peak annotation. An in-house MS2 database was applied to identify the metabolites. We obtained 1918 annotated metabolites and 1016 unknown analytes in the positive and negative ion model, and chose the annotated metabolites for subsequent analysis.

The partial least square discriminant analysis (PLSDA) of the metabolomics data was conducted by the mixOmics R package.

### 2.4. l-DNB Analysis

The l-DNB method we used for detecting the individual early-warning signals in complex disease was reported previously [14] with slight modification in this study.

Briefly, the DNB identification using the l-DNB method was based on the three criteria of previous DNB theory [11]. As shown in Figure 2a, when a biological system approaches the predisease state from a normal state, a group of biomolecules (i.e., DNB module) satisfies the following three statistical conditions [13,15,16,17]:
The deviation of each molecule inside the module ($SD_{in}$, standard deviation) fluctuate strongly;The correlation among molecules inside the module ($PCC_{in}$, Pearson correlation coefficients in absolute values) dramatically increases; andThe correlation of the molecules between the inside and outside of this module ($PCC_{out}$, Pearson correlation coefficients in absolute values) dramatically decreases.


(1) Construction of SSNs

In order to calculate the DNB at individual level, we need to construct a single-sample network (SSN) for the given samples first. The theoretical principle of the SSN method has been reported before [18]. Briefly, based on a group of reference samples (*n* samples), a reference network can be constructed by the correlations between molecules using the abundance data of this group. The PCC between molecules *x* and *y* in the data of the *n* reference samples (i.e., samples in control group) can be calculated as
(1)$$PCC_{n}\left(x,\;y\right)=\frac{\sum_{i}^{n}\left(x_{i}-{\bar{x}}_{n}\right)\left(y_{i}-{\bar{y}}_{n}\right)}{\sqrt{\sum_{i=1}^{n}{\left(x_{i}-{\bar{x}}_{n}\right)}^{2}}\sqrt{\sum_{i=1}^{n}{\left(y_{i}-{\bar{y}}_{n}\right)}^{2}}},$$
where $x_{i}$ and $y_{i}$ are the values of molecules x and y in the *i*th reference sample among the *n* reference samples, respectively. And ${\bar{x}}_{n}$ and ${\bar{y}}_{n}$ are the respective average values for molecules *x* and *y* in the *n* reference samples. 

When one new sample d is added to the *n* reference samples, the PCC between molecules *x* and *y* in the data of (*n*+1) samples (one new sample + the original *n* reference samples) can be calculated as
(2)$$PCC_{n+1}\left(x,\;y\right)=\frac{\sum_{i}^{n+1}\left(x_{i}-{\bar{x}}_{n+1}\right)\left(y_{i}-{\bar{y}}_{n+1}\right)}{\sqrt{\sum_{i=1}^{n+1}{\left(x_{i}-{\bar{x}}_{n+1}\right)}^{2}}\sqrt{\sum_{i=1}^{n+1}{\left(y_{i}-{\bar{y}}_{n+1}\right)}^{2}}},$$
where $x_{i}$ and $y_{i}$ are the values of molecules x and y in the *i*th sample among the (*n* + 1) samples, respectively. And ${\bar{x}}_{n+1}$ and ${\bar{y}}_{n+1}$ are the respective average values for molecules *x* and *y* in the (*n* + 1) samples. 

The influence of the new sample d is mainly reflected in the changes of PCC. Therefore, the differential PCC between molecules *x* and *y* for sample d against the *n* reference samples, is defined as
(3)$$sPCC_{n}\left(x,\;y\right)=PCC_{n+1}\left(x,\;y\right)-PCC_{n}\left(x,\;y\right).$$
The $sPCC_{n}\left(x,\;y\right)$ follows the volcano distribution [18], which approximates a normal distribution when *n* is large enough. Thus, we can use a statistical hypothesis test (*Z*-test or *U*-test) to evaluate whether the molecules *x* and *y* are significantly correlated at the single-sample level. If the *p*-value < 0.05, *x* and *y* are considered to have significant correlation.

(2) Calculating the local DNB score for each molecule in a single sample

Based on the sample-specific network, we defined the target molecule and its first-order neighbors as a local module. According to the three statistical conditions of DNB theory, the local DNB score for each molecule in sample d can be defined as follows:(4)$$I_{s}\left(x\right)=sAD_{in}\frac{sPCC_{in}}{sPCC_{out}},$$
where $I_{s}\left(x\right)$ is the score of the local module of molecule *x* in the single sample *d*. The variables in the equation are calculated as follows:

Corresponding to the $SD_{in}$ in DNB theory, we define the single-sample Abundance Deviation (sAD) as follows:(5)$$sAD\left(x_{d}\right)=|x_{d}-\bar{x}|,$$
where $x_{d}$ is the abundance of molecule *x* in the new sample *d* and $\bar{x}$ is the average abundance of molecule *x* in the *n* reference samples. Then, we infer the following relational expression: (6)$$sAD_{in}=\frac{1}{1+n_{x_{d}}}\left[sAD\left(x_{d}\right)+\underset{y_{d}\epsilon N_{x_{d}}}{\sum }sAD\left(y_{d}\right)\right],$$
which represents the average differential deviation in abundance of all $\left(1+n_{x_{d}}\right)$ molecules in the local module of molecule *x* for sample d against the n reference samples. Furthermore, $N_{x_{d}}$ is a molecule set with $n_{x_{d}}$ molecules, which is the first-order neighbors of the molecule x. 

Next, we calculated the correlation between molecules within the module and the correlations between inside and outside of the module, which were named as $sPCC_{in}$ and $sPCC_{out}$, respectively. The $PCC_{in}$ is defined as
(7)$$sPCC_{in}=\frac{1}{n_{x_{d}}}\underset{y_{d}\epsilon N_{x_{d}}}{\sum }sPCC_{n}\left(x_{d},\;y_{d}\right),$$
where $sPCC_{in}$ is the average value of $sPCC_{n}$ between molecule $x_{d}$ and all its first-order neighbors, $N_{xd}$. Furthermore,
(8)$$sPCC_{out}=\frac{1}{n_{x_{d}}m_{x_{d}}}\underset{\begin{array}{c} y_{d}\epsilon N_{x_{d}}\\ z_{d}\epsilon M_{x_{d}} \end{array}}{\sum }sPCC_{n}\left(y_{d},\;z_{d}\right),$$
where $sPCC_{out}$ is the average of $sPCC_{n}$ between the first-order neighbors and second-order neighbors of molecule $x_{d}$. $M_{x_{d}}$ is a molecule set with $m_{x_{d}}$ molecules, which is the number of second-order neighbors of molecule $x_{d}$. All the above modules and molecules are based on SSN.

It is worth noting that we only considered the molecules that have at least three first-order neighbors and one second-order neighbor in the network topology. The first-order neighbors and the second-order neighbors are disjoint molecular sets. Additionally, when calculating the DNB scores of the control group, we used the same reference network as the Stage A and B groups.

### 2.5. Pathway Enrichment of l-DNB Molecules and Related Genes

The pathway enrichment analysis for l-DNB molecules was performed by MetaboAnalystR 3.0 R package. The canonical pathway analysis of the l-DNB molecule-related genes was carried out by Ingenuity Pathway Analysis (IPA) software (QIAGEN^®^ Bioinformatics, Germantown, MD, USA).

### 2.6. Heatmap of Local DNB Molecule Related Genes

The heatmap of local DNB molecule-related genes was conducted by the gplots R package. The gene expression data used in the heatmap were RNA-Seq data of peripheral blood mononuclear cells from the patients that received selective coronary angiography between June 2011 and March 2015 in Fuwai Hospital. Twenty-four samples (8 samples per group by random sampling) were sequenced on an Illumina platform with paired end 150 bp. The clean data filtered from raw data were aligned to the Human reference genome ENSEMBL GRCh38.p13 using HISAT2 program, and then assembled and quantified by using StingTie software. The average log2 fold change value between groups was analyzed by DESeq2 R package. 

### 2.7. Statistical Analysis

Continuous data were expressed as mean ± SD and compared using the ANOVA test. Categorical data were expressed as count (percentage) and compared using the chi-square or Fisher exact test. *p*-Value < 0.05 was considered statistically significant. The statistical result of global DNB scores from all samples is presented as mean ± SEM using GraphPad Prism 5.0 software (GraphPad Software Inc., San Diego, CA, USA) (Figure 3a). The bar plots and line chart of statistical results in this study are displayed by GraphPad Prism 5.0 software as well (Figure 3b,c). 

## 3. Results

### 3.1. Global Plasma Metabolomics Profile of Coronary Atherosclerosis Patients

To detect the early-warning signals during the progression of human coronary atherosclerosis, the plasma was collected from patients with different pathological states. As shown in Figure 1a, according to the coronary angiography results of patients, the samples were generally divided into three groups—control group (the stenosis of coronary artery is less than 20%), Stage A group (the stenosis of coronary artery is between 50% and 70%) and Stage B group (diagnosed as AMI). The baseline characteristics are shown in Table 1. There was no significant difference among the three groups in baseline characteristics. To get the global metabolomics profile during the progression of coronary atherosclerosis, these 48 plasma samples were detected on Ultra High-Performance Liquid Tandem Chromatography Quadrupole Time of Flight Mass Spectrometry (UHPLC-QTOFMS). After a series of data preprocessing (Figure 1a), we finally identified about 1900 annotated metabolites by relative quantification, including both positive and negative ion models. Through the partial least squares discriminant analysis (PLSDA) result of metabolomics data (Figure 1b), we found that Stage A group was quite different from the other two groups in terms of metabolites, which indicated that there might exist a tipping point during the nonlinear process of coronary atherosclerosis progression. 

### 3.2. The Tipping Point of Each Individual during the Progression of Human Coronary Atherosclerosis by l-DNB

The metabolomics profile and previous evidence suggested that the progression of human coronary atherosclerosis is a nonlinear process, and there exists a tipping point during the disease development. The traditional methods based on differentially expressed genes are limited by their static characteristics, which fails to distinguish the predisease state from the normal state. Thus, we turn to the DNB theory (Figure 2a), which was developed for identifying the tipping point of complex diseases and discovering the critical networks [11,15,19]. When the system approaches the tipping point, a dominant group of molecules, named DNBs, satisfies the three statistical conditions (i.e., strong fluctuation, high correlation of internal molecules and low correlation with external molecules) (See Methods—l-DNB analysis). Based on these, it is available for us to identify the tipping point of human coronary atherosclerosis and discover the potential network biomarkers. However, even with the same pathological phenotype, there are great differences among individuals. Moreover, the DNB method identifies the tipping point of the whole system by using multiple samples, which is unavailable for personal prediction. 

To solve these problems, the new method, l-DNB, was developed (Figure 2b). First, by using the metabolomics data, we constructed a single-sample network (SSN) for given samples based on the reference samples (samples in the control group). Next, according to the three criteria of DNB theory, we calculated the local DNB score of each molecule in each sample. Finally, the molecules in each sample were sorted by their local DNB score, and the global DNB score for each sample was the mean value of all local DNB scores in this sample. The detailed information of the calculation can be found in Methods. 

According to the above calculation, we obtained a landscape of local DNB molecule scores for each sample. As shown in Figure 3a, the average value of global DNB scores in the Stage A group was significantly higher than that in other two groups, which means that the tipping point of human coronary atherosclerosis is at Stage A. This result coincides with the critical stage defined in clinical medicine. Then, we defined the person with a global DNB score greater than 1.0 as the patient at high risk of adverse cardiovascular events. We found that most of the patients (13 out of 15 persons) in the Stage A group were accurately predicted to be the high-risk patients, which means the true positive rate (TPR) is 86.76% (Figure 3b). Meanwhile, the predicted high-risk patients in the Control group and Stage B group were only five and two persons, respectively. Thus, the false positive rate (FPR) is only 21.21%. The global DNB score of each sample can be found in Appendix A. These results demonstrated that the l-DNB method can accurately identify the patient who is at the critical state (tipping point).

Although the patients in the Stage A group received the corresponding treatment after the coronary angiography, the patient (ID: XJ608) still underwent major cardiovascular events according to the follow-up information collected in August 2019. Therefore, taking the XJ608 patient for example to test the effectiveness and accuracy of our analysis results, we evaluated the composition of local DNB molecules and the robustness of prediction. First, we respectively selected the top five (or top 10, top 15, top 20, …, top 500, i.e., total 100 times) local DNB molecules to calculate the global DNB score of each sample. In this way, every sample had 100 global DNB scores corresponding to different molecular sample sizes. Then, we compared the global DNB score of “XJ608” patient with the corresponding global DNB scores of other samples, and finally got the ranking distribution of “XJ608” in 100 times of such sampling. The statistic result of the sample “XJ608” ranking showed that the probability of sample “XJ608” ranking in the top three relative to all samples is 0.52 and the probability of ranking in top five is 0.91 (Figure 3c), which implied that XJ608 patient had a high risk of morbidity. This result further confirms that the l-DNB method can personalize the tipping point of coronary atherosclerosis accurately.

To further investigate the mechanism of molecule networking that drives the system as it approaches the tipping point, we selected the molecules with local DNB scores in the top 50 from each sample of the Stage A group, and these molecules were also included in at least three samples of the Stage A group. Through the pathway enrichment analysis of these DNB molecules (Figure 3d), we found that these molecules were mainly enriched in pathways related to lipid metabolism, fatty acid metabolism and amino acid metabolism, which implied that these enriched metabolic pathways played important roles at the tipping point of human coronary atherosclerosis.

Among the DNB molecules mentioned above, 16 l-DNB molecules were candidate molecules (top 50) in at least half of the Stage A samples (Figure 4a). In addition, these molecules include many phospholipids and fatty acids. Therefore, we extracted the related genes involved in the metabolism of these molecules using the KEGG database, and obtained their dynamic expression in the three stages using the RNA-Seq data of peripheral blood mononuclear cells in coronary atherosclerotic patients (Appendix A). Through the dynamic expression of the l-DNB molecule-related genes, we found that most of the genes reversed their expression trend after the tipping point (Figure 4b). Moreover, the canonical pathway enrichment results of these genes showed that most of the enriched pathways were activated before the tipping point (Stage A group compared with the Control group), except one pathway (antioxidant Action of Vitamin C) that was suppressed (Figure 4c). After the tipping point (Stage A group compared with Stage B group), the states of these pathways were reversed. These results suggested that lipid metabolism-related genes and molecules played the key roles in driving the coronary atherosclerosis approaching the tipping point.

## 4. Discussion

The progression of human coronary atherosclerosis is a nonlinear dynamic process, which is influenced by many factors, such as heredity, environment and living habits, etc. [20,21]. Due to the complexity of the disease, patients at a high risk of adverse cardiovascular events cannot be effectively identified by the degree of vascular stenosis [22]. Therefore, we must develop new methods for the prediction and diagnosis of an individual patient. The DNB method is able to detect the critical states during disease progression or biological processes, and its effectiveness has been studied in many papers from both experimental and computational viewpoints [12,13,23,24,25], e.g., cell fate decision [23,24], immune checkpoint blockade [25], pulmonary metastasis [13] and steatohepatitis transition [12]. 

In this study, using an untargeted metabolomics approach, we not only accurately identified the tipping point during the progression of human coronary atherosclerosis, but also precisely assessed the risk of adverse events on individual patients by using the l-DNB method.

The metabolomics profiles of plasma enable us to systematically study the mechanism of molecular dynamics in pathogenesis under noninvasive conditions, which are widely used to identify novel biomarkers in the cardiovascular field [26,27]. Using the plasma metabolomics data at different stages, we identified that the tipping point during the progression of human coronary atherosclerosis is at Stage A, which corresponds to the critical stage defined in clinical medicine [5,6]. More importantly, we accurately identified the early-warning signals for individual patients by their global DNB scores with the TPR of prediction 86.67%, and the FPR only 21.21%. Due to limited sample size, there was only one patient (ID: XJ608) who underwent a major cardiovascular event after medication in the Stage A group. However, in the ranking of patients that were predicted to be at high risk of adverse events, the probability that the XJ608 patient ranked in the top five in 100 times of sampling was 0.91. This result also strongly proves that the l-DNB method can accurately personalize the early-warning signals of coronary atherosclerotic patients.

Meanwhile, we discovered a group of DNB molecules that may play driving roles in the molecular regulatory networks during the progression of coronary atherosclerosis. Many of them have been reported to be involved in lipid metabolism and fatty acid metabolism, which validates our results. Phosphatidylglycerol (PG) and phosphatidylethanolamine (PE), the subclasses of glycerophospholipids, have been reported to be observed in plasma lipoproteins and atherosclerotic plaques [28]. Depending on the species of the bound fatty acids, different kinds of glycerophospholipids play different roles in biological processes [28,29]. Although some kinds of PEs have been reported to be closely involved in the progression of atherosclerosis [30,31], the functions of other kinds of PEs remain concealed. Current studies demonstrate that PGs are the important participants in the signal transduction and stress response [32], but their roles in atherosclerotic progression are still unclear. In our result, PG (16:1/16:1) and PG (18:3/20:4) are DNB molecule candidates in 14 samples of the Stage A group (15 samples in total), which reveals the importance of PGs in the progression of coronary atherosclerosis, and their specific molecular mechanisms need to be further studied. Palmitic acid, a molecule involved in fatty acid metabolism, has been reported to be intimately related to cardiovascular disease [29,33,34]. The metabolic dynamic changes caused by palmitic acid intake can influence the levels of total cholesterol and low-density lipoprotein (LDL) cholesterol, which show a strong positive correlation with the development of coronary heart disease [35,36,37]. Our study revealed that palmitic acid is an important DNB molecule, and its dynamic fluctuation as well as the participating regulatory network can drive the progression of coronary atherosclerosis. However, the underlying molecular mechanism needs to be further investigated. More importantly, for the first time, we took these metabolites as a network biomarker (by using their second-order statistics for disease prediction, e.g., correlation and deviation of these molecules). Although these metabolites have been studied separately, taking them as a network (evaluating them as a network, e.g., not by the traditional first-order statistics but by the second-order statistics) to assess the disease status will provide new insights into the discovery of novel network biomarkers.

In conclusion, based on the l-DNB method, our study identified the tipping point of human coronary atherosclerosis at Stage A, which was consistent with the clinically defined critical stage. More importantly, we detected the early-warning signal of each patient and accurately identified the patients at high-risk of adverse events. These not only provide a new noninvasive method for the early prediction of adverse cardiovascular events, but also contribute to the realization of personalized diagnosis. Furthermore, we also discovered a group of DNB molecules that play driving roles in the development of the disease, which provide a new insight into the molecular network mechanisms of human coronary atherosclerosis. 

## Figures and Tables

**Figure 1 genes-11-00676-f001:**
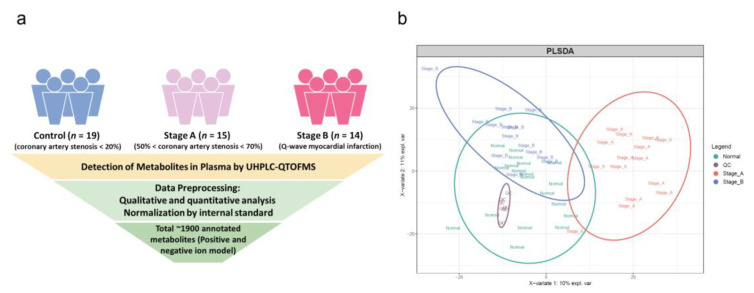
(**a**) The schematic diagram of study design; (**b**) partial least squares discriminant analysis (PLSDA) of global plasma metabolomics profile.

**Figure 2 genes-11-00676-f002:**
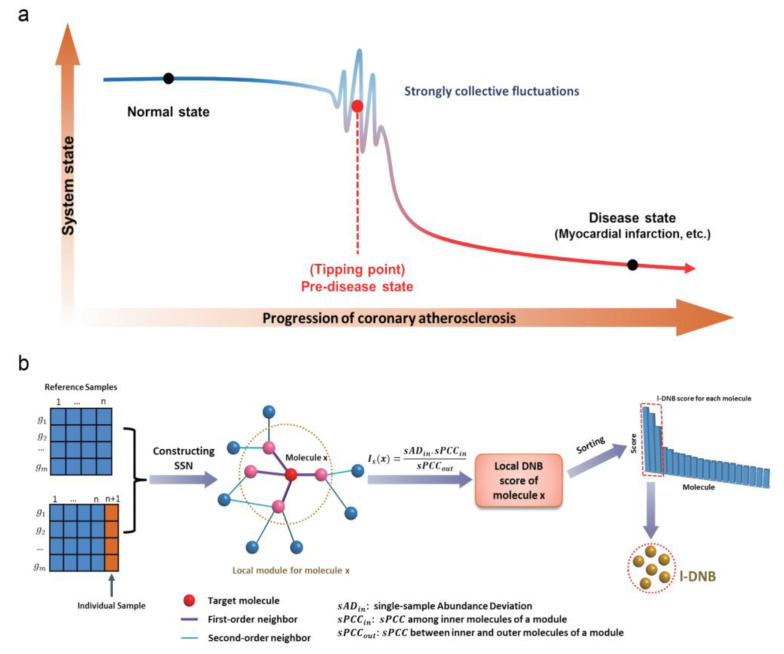
(**a**) The schematic diagram for the dynamical progression of human coronary atherosclerosis. The process of disease development can be divided into three states—normal state, predisease state and disease state. The system is stable and ordered at a normal state. With evolution of the disease, the system reaches the predisease state (i.e., tipping point), which has no substantial changes in pathological phenotype compared with the normal state, and it is reversible between the predisease state and normal state. However, when the system crosses the tipping point, it deteriorates rapidly, eventually leading to the major cardiovascular events. (**b**) The flowchart of the landscape dynamic network biomarkers (l-DNB) method to identify DNB molecules from a single sample.

**Figure 3 genes-11-00676-f003:**
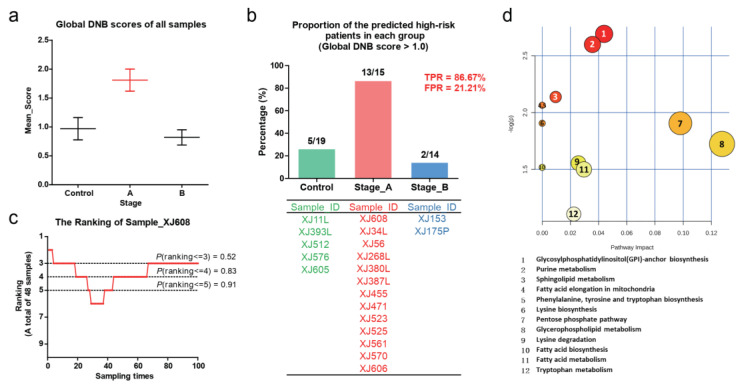
(**a**) The statistical results of the global DNB scores in each group indicated that the tipping point of coronary atherosclerosis is at Stage A. (**b**) The proportion of the predicted high-risk patients (global DNB score > 1.0) in each group (upper panel). Furthermore, the IDs of these patients in each group were listed below. The true positive rate (TPR) is 86.67% and false positive rate (FPR) is 21.21%. (**c**) The ranking of the patient (XJ608) in sampling 100 times. (**d**) The pathway enrichment of DNB molecule candidates. The color is used to distinguish different groups.

**Figure 4 genes-11-00676-f004:**
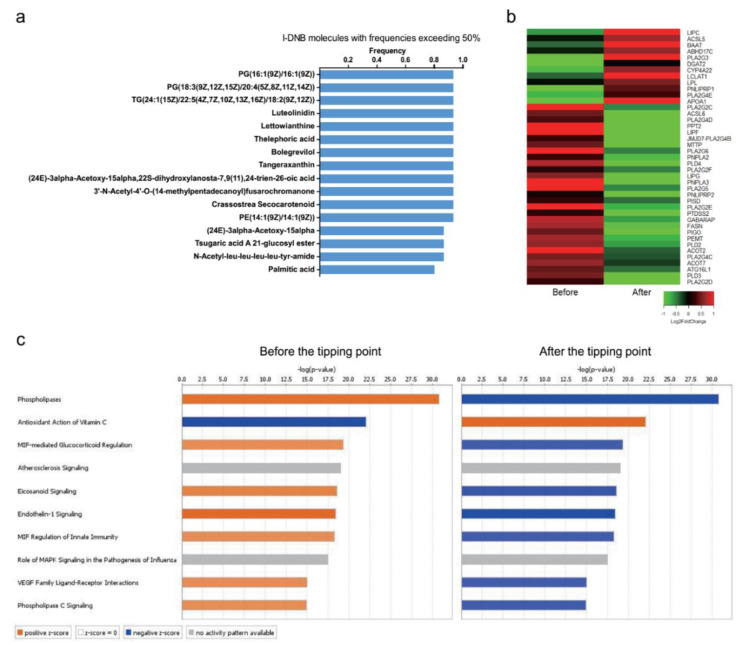
(**a**) The list of l-DNB candidate molecules with frequencies exceeding 50% in Stage A samples. (**b**) The dynamic expression of the l-DNB candidate molecule-related genes before and after the tipping point. Before the tipping point—the average log2 fold change of Stage A group relative to Control group. After the tipping point—the average log2 fold change of Stage A group relative to Stage B group. (**c**) The canonical pathway analysis of the l-DNB candidate molecule related genes. The top 10 enriched pathways are shown.

**Table 1 genes-11-00676-t001:** Baseline Characteristics.

	Control *n* = 19	Stage A *n* = 15	Stage B *n* = 14	*p*-Value
Age (years)	58.53 ± 8.64	61.38 ± 9.68	61.50 ± 9.35	0.563
Female	9 (47.4%)	7 (43.8%)	7 (50%)	0.942
BMI (kg/m^2^)	27.22 ± 7.95	25.12 ± 4.46	25.99 ± 2.49	0.558
SBP (mmHg)	126.32 ± 12.11	126.13 ± 13.29	123.86 ± 16.43	0.863
DBP (mmHg)	78.95 ± 11.97	73.44 ± 7.69	75.50 ± 11.33	0.305
Diabetes mellitus	3 (15.8%)	2 (12.5%)	5 (35.7%)	0.236
Hypertension	11 (57.9%)	12 (75%)	10 (71.4%)	0.521
Dyslipidemia	10 (52.6%)	12 (75%)	12 (85.7%)	0.207
Family history of CAD	3 (15.8%)	6 (37.5%)	4 (28.6%)	0.343
Premature CAD	2 (10.5%)	4 (25%)	1 (7.1%)	0.316
Cerebrovascular disease	0 (0.0%)	1 (6.3%)	4 (28.6%)	0.061
Peripheral vascular disease	3 (15.8%)	3 (18.8%)	1 (7.1%)	0.638

BMI: body mass index; SBP: systolic blood pressure; DBP: diastolic blood pressure; CAD: coronary artery disease.

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
