# Peer review of "Personalized Early-Warning Signals during Progression of Human Coronary Atherosclerosis by Landscape Dynamic Network Biomarker"

_genes, 2020, doi:10.3390/genes11060676_

Round 1

Reviewer 1 Report

The authors suggested an improved method for detecting tipping point of dynamical system and applied it to a human disease coronary atherosclerosis. They found that the dynamical network biomarker (DNB) score can be a good indicator of early-warning signals for detecting high-risk pre-disease patients. Furthermore, the RNA-seq and pathway analysis showed that the global gene expression related to lipid metabolism is changed after tipping point. This new technology is worth to be used for detecting tipping point of other human diseases. However, the paper has some critical point to improve.

  1. The authors developed landscape-DNB (I-DNB) by combining DNB and single-sample network (SSN) which are published before by the same research group. The previous methods DNB was also developed for detecting tipping points. However, the author did not explain why the new method I-DNB was needed and how I-DNB is better than DNB in terms of detecting tipping points.
  2. When the authors compared the DNB scores of control, group A, and group B, they used the mean value of all DNB scores. On the other hand, when the authors ranked the DNB scores of XJ608, they used the mean value of top 5 DNB scores. This difference of analysis strategy should be explained.
  3. In the line 262, the author mentioned "these steps were repeated 100 times". It is not clear that for which samplings the steps were repeated. Was the top 5 DNB scores randomly sampled?
  4. The samples of RNA-seq data is not clearly explained. What do you mean "Before" and "After" the tipping point? Is "Before" and "After" means sample A and B? In method, the author mentioned that twenty-four samples were sequenced. Does it mean that the author obtained twenty-four RNA-seq data? Or the samples from the same group were pooled for sequencing?
  5. The criteria of identifying SSN such as p-value criteria is not described in the method section.
  6. The sPCCout is calculated by the average of sPCC between the first-order neighbors and the second-order neighbors. Can the members between the first-order and second order neighbors be shared? Or are they disjoint sets?
  7. The DNB score is calculated for each sample by comparing its PCC values with the PCC values of the control group. Then, how was the DNB scores for control samples calculated? When the DNB scores for a sample in control group, was the sample excluded and was every other samples in the control group considered as control samples?
  8. Some figures have very poor resolution such as Figure1b, Figure 4c.
  9. The abbreviation of single-sample Abundance Deviation in the methods section is sAD(xd) but it is sEDin in Figure 2b. It looks like that there is typo in Figure 2. "Single-simple" should be "Single-sample".
  10. The reason why the progression of human coronary atherosclerosis is a non-linear process was not clear. Is there any reference? Or is there any non-linearity in the metabolomics data? If the non-linearity was found in the data, it should be shown.

Reviewer 2 Report

Ge et al. investigated the landscape dynamic network biomarkers methodology to identify the early-warning signal of human coronary atherosclerosis. Some study concerns should be mentioned.

  1. As for global metabolomics measurement, the author should provide the information of known (annotated) metabolites and unknown analytes detected in this study.
  2. The study aims to evaluate the dynamical progression of human coronary atherosclerosis and try to detect the pathological signaling for the disease processing early. However, The data were collected from different individuals stratified by disease status. Thus, how to evaluate the signaling from inter-individual differences rather than the disease status difference. To systematic approaches the tipping point, a panel data within one person is better to catch the early signaling. Therefore, the tipping point of each individual during the progression of human coronary atherosclerosis is misleading.
  3. In this study, three groups were allocated as a control group (the stenosis of the coronary artery is less than 20%), Stage A group (the stenosis of the coronary artery is between 50% and 70%), and Stage B group (diagnosed as AMI). However, misclassification of candidate metabolites could be noted in this 3-group stratification. According to the classification of luminal stenosis of coronary diseases. Several reports suggested the luminal stenosis of coronary diseases was classically categorized into 4 degrees based on the percentage of cross-sectional area stenosis: 1% to 25%, 26% to 50%, 51% to 75%, and 76% to 100% [1–3]. Thus, if we divided the samples into four groups, such as 1% to 25%, 26% to 50%, 51% to 75%, and 76% to 100%, would the metabolite signaling be the same or not?

Reference

[1]. Scanlon PJ, et al. ACC/AHA guidelines for coronary angiography: executive summary and recommendations. A report of the American College of Cardiology/American Heart Association Task Force on Practice Guidelines (Committee on Coronary Angiography) developed in collaboration with the Society for Cardiac Angiography and Interventions. Circulation. 1999; 99:2345–2357.

[2]. Baim DS, et al. Problems in the evaluation of new devices for coronary intervention: what have we learned since 1989?Am J Cardiol. 1997; 80(10A):3K–9K.

[3]. Chen X, et al. A pathological study of sudden coronary death in China: report of 89 autopsy cases. Forensic Sci Int. 1992; 57:129–137.

  1. In the results, the enrichment analysis of DNB molecules indicated lipid metabolism, fatty acid metabolism, and amino acid metabolism were important for the tipping point of human coronary atherosclerosis. This information had been well-established in clinical practice. What is the novel finding using the DNB method to detect the process of human coronary atherosclerosis?
  2. How to select the eight samples per group to evaluate the RNA-Seq results? By random sampling or other methods? The author should mention in their method part.
  3. In the discussion section, the author should emphasize their clinical implication using the l-DNB method to identify the candidate metabolites and address the importance of the selected metabolites as candidate biomarkers. The author mentioned that this is a new non-invasive method for early prediction of adverse cardiovascular events. However, I didn’t see a better prediction using this method compared to traditional methodology or other methodologies.

Reviewer 3 Report

In this manuscript titled, "Personalized early-warning signals during the progression of human coronary atherosclerosis by landscape dynamic network biomarker", Jing Ge et al., authors investigate the tipping point (pre-disease state of disease) and detecting early-warning signals of human coronary atherosclerosis that based on the l-DNB methodology, by using the metabolomics data of plasma of patients with coronary atherosclerosis at different stages. This manuscript is written clearly, however, the manuscript appears preliminary.

  1. Authors should improve the quality of figure 1b, it’s very blurry.
  2. As shown in figure 2a, the progression of coronary atherosclerosis contains three stages. In order to identify the early-warning signals during the progression of human coronary atherosclerosis, the normal human control group should be considered.
  3. The authors discovered a group of DNB molecules, which may play driving roles in the molecular regulatory networks during the progression of coronary atherosclerosis. The authors only predicted the targets but still need to validate them.

Round 2

Reviewer 1 Report

The author properly revised the manuscript to meet most of the comments. I only don't understand one point.

As I mentioned, the way of calculating DNB scores for control group is still not clear. The sPCC is calculated by the difference between PCC of "reference samples + one new sample" and PCC of "reference sample". Let say we calculate sPCC of one sample in control group. Then the sPCC should be 0 because the PCC of "reference samples + one new sample" is same with the PCC of "reference samples" (one new sample is already included in the reference samples". But in the Figure 3a, the global DNB scores is about 1 with some value of variance.

Reviewer 2 Report

All questions and comments had been addressed. I have no other suggestion.

Author Response

Thank you for the valuable comments and suggestions.

Reviewer 3 Report

The actual downstream mechanism of regulatory networks is very important to support this study. Currently, the data were not enough. In the author’s reply, they are already working on it, so when they have the biological experiment data to verify their predicted results, this study might be more acceptable at that time.
